# Sleep deprivation did not enhance the success rate of chloral hydrate sedation for non-invasive procedural sedation in pediatric patients

Yu Cui⬤*◐, Langtao Guo◐, Qixia Mu, Qin Cheng, Lu Kang, Yani He, Min Tang, Qunying Wu

Department of Anesthesiology, The Affiliated Hospital, School of Medicine, UESTC Chengdu Women's & Children's Central Hospital, China

◐ These authors contributed equally to this work.
* cuiyu19831001@163.com

**Data Availability Statement:** The data are all contained within the manuscript.

**Funding:** The authors received no specific funding for this work.

## Abstract

### Study objective

In Asian countries, oral chloral hydrate is the most commonly used sedative for non-invasive procedures. Theoretically, mild sleep deprivation could be considered as one of assisted techniques. However, there is no consensus on sleep deprivation facilitating the sedation during non-painful procedures in children. The aim of our study is to analyze the clinical data of children undergoing non-invasive procedural sedation retrospectively and to evaluate the association between mild sleep deprivation and sedative effects in non-invasive procedures.

### Measurements

Consecutive patients undergoing chloral hydrate sedation for non-invasive procedures between December 1, 2019 to June 30, 2020 were included in this study. The propensity score analysis with 1: 1 ratio was used to match the baseline variables between patients with sleep deprivation and non-sleep deprivation. The primary outcome was the failure rate of sedation with the initial dose. The secondary outcomes included the failure rate of sedation after supplementation of chloral hydrate, the incidence of major and minor adverse events, initial and supplemental dose of chloral hydrate, and the length of sedation time.

### Main results

Of the 7789 patients undergoing chloral hydrate sedation, 6352 were treated with sleep deprivation and 1437 with non-sleep deprivation. After propensity score matching, 1437 pairs were produced. The failure rate of sedation with initial chlorate hydrate was not significantly different in two groups (8.6% [123/1437] vs. 10.6% [152/1437], p = 0.08), nor were the failure rates with supplemental chlorate hydrate (0.8% [12/1437] vs. 0.9% [13/1437], p = 1) and the length of sedation time (58 [45, 75] vs. 58 [45, 75] min; p = 0.93).

**Competing interests:** No authors have competing interests.

## Conclusions

The current results do not support sleep deprivation have a beneficial effect in reducing the pediatric chloral hydrate sedation failure rate. The routine use of sleep deprivation for pediatric sedation is unnecessary.

## Introduction

To our knowledge, some pediatric patients cannot cooperate with some non-invasive procedures, including MRI, CT, cardiac ultrasound, and lung function. During those procedures, to obtain high quality of examinations, sedation is needed to provide both immobilized and sleep states. The clinical practice statement of the European Academy of Pediatric Anesthesiology has clarified the definition of three different target sedation states (minimal, moderate, and deep sedation) [1]. The goal of sedation is to provide the safety of the children while ensuring the sedation effects. In clinical practice, the sedation techniques are divided into drug and non-drug techniques. The choice of sedation techniques depends upon the specialists' assessment. Among the drug techniques, the most widely used are sedative-hypnotics, including midazolam, chloral hydrate, dexmedetomidine and propofol [2]. For patients without intravenous access undergoing non-painful procedure (i.e., diagnostic imaging), oral or transmucosal routes are more convenient and less invasive. Midazolam, chloral hydrate, and dexmedetomidine that can be administrated orally or transmucosally are considered as the best choices. However, sedative-hypnotics, such as midazolam, chloral hydrate, dexmedetomidine, have a certain failure rate about 1% [3], 3.4% [4], 5.7% [5], respectively. Failure of sedation may lead to cost and time detriment, as well as dissatisfaction from guardians. Besides, the narrow margin of drug effectiveness and toxicity requires skills, knowledge, and experience. High dose of dexmedetomidine may cause bradycardia [5]. Additionally, midazolam syrup is not commercially available in China, which limits its widely used. To the best of our knowledge, in Asian countries, oral chloral hydrate is the most commonly used medication for diagnostic imaging [6,7].

Recently, the non-drug techniques were highly recommended by the European Society for Pediatric Anesthesiology, especially in non-invasive procedural sedation [1]. Theoretically, mild sleep deprivation could be considered as one of the non-drug techniques. Valenzuela et al. had performed a clinical study about chloral hydrate sedation for auditory brainstem response (ABR) testing, and the authors found that the most common reason for sedation failure was lack of sleep deprivation of the children [4]. Compared with natural sleep, following sleep deprivation infants maintained a greater proportion of quiet sleep, but without interruption of arousal propensity [8]. On the contrary, a retrospective study presented that sleep deprivation had no effect in reducing the pediatric sedation failure rate [9]. Thus, there was no consensus on sleep deprivation facilitating the sedation during non-painful procedures in children.

At our clinical children sedation unit, mild sleep deprivation was routinely recommended before sedation. We therefore analyzed the clinical data of children retrospectively, and our primary purpose was to evaluate the association between mild sleep deprivation and sedative effects during non-invasive procedures.

## Methods

### Data collection

This study was approved by Institutional Review Board (IRB) of Chengdu Women's and Children's central Hospital and was registered at http://www.chictr.org.cn/index.aspx with No.

ChiCTR2000036126. The patient data from December 1, 2019 (from the very beginning of electrical medical record utilization) to June 30, 2020 was pooled. Considering the nature of the retrospective study and the anonymized patient data, the necessity of informed consents for data published was waived.

In our hospital, the moderate-to-deep sedation unit is responsible for the management of sedation, which serves pediatric population, both inpatient and outpatient. The responsibilities of this unit only provide moderate or deep sedation for non-invasive procedures. Any painful procedures are performed at the anesthesia surgery center and are not included in the study.

The eligible criteria were as follows: (1) children who underwent drug assistant sedation for non-invasive procedures in our sedation center; (2) pediatric patients who received choral hydrate as the initial and supplemental sedative; The exclusion criteria were as follows: (1) incomplete data; (2) age $\leq$ 28 days; (3) patients on assisted ventilation. The retrospective data, including patients' age, gender, weight, diagnosis, allergies, history of sedation, history of sedation failure, types of procedure, initial and supplemental dose of choral hydrate, sleep deprivation and adverse events, the length of sedation, were collected.

## Sedation method

In our institution, to decrease the incidence of sedation failure, for appointed patients, mild sleep deprivation is encouraged the night prior to the procedure, though for some children it is not a practical option. Mild sleep deprivation is defined as having children wake up 2–4 hours earlier. For non-appointed patients, sleep deprivation cannot be achieved. On arrival to the sedation unit, the guardians are asked in a nonjudgmental question (e.g., How many hours of sleep their child received yesterday? And how many hours of sleep their child usually received?). Any child who sleeps more than 2 hours less than usual is considered sleep deprived. In addition, if the child takes a nap on the way to the hospital, they are considered non-sleep deprived. Then, the children were divided into sleep deprivation plus choral hydrate group (sleep deprivation group) and choral hydrate with non-sleep deprivation (non-sleep deprivation group).

An experienced pediatric anesthesiologist (>2 years) and six nurses who have completed a training on the safe pediatric procedural sedation are in charge of sedation on each business day. Patients are fasted from clear liquids for 2h and from formula, breast milk for 4h per our institutional guidelines. The clear liquids were referred to water, fruit juices without pulp, or carbohydrate-containing fluids without protein. The pre-sedation evaluation and the acquisition of written informed consent are the responsibility of the pediatric anesthesiologist. Then, the pediatric anesthesiologist is responsible for prescribing medications, including the type, dose, and route. The choral hydrate is routinely used for non-painful diagnostic procedure sedation. In our study, only patients who received prescription of chloral hydrate as the initial and supplemental sedative are included. If the initially dose of choral hydrate is not meet the individual requirements, supplemental dose will be prescribed. The type and route of supplemental medication will be determined by the pediatric anesthesiologist based on the assessment.

After the assessment, the patients are closely monitored, and the heart rate and $SPO_2$ are recorded. One of nurses prepares the chloral hydrate according to anesthesiologist's prescription. In our unit, as the majority of patients undergoing non-painful procedures do not have intravenous access, oral chloral hydrate is preferred. Based on previous study [2], chloral hydrate is administered at a dose of 25–100 mg/kg by the oral or rectal route for patients, supplemental dose of 50 mg/kg after 30 min, and at a maximum total dose of 2 g or 100 mg/kg (whichever is less). In clinical practice, considering individual differences and patients' safety,

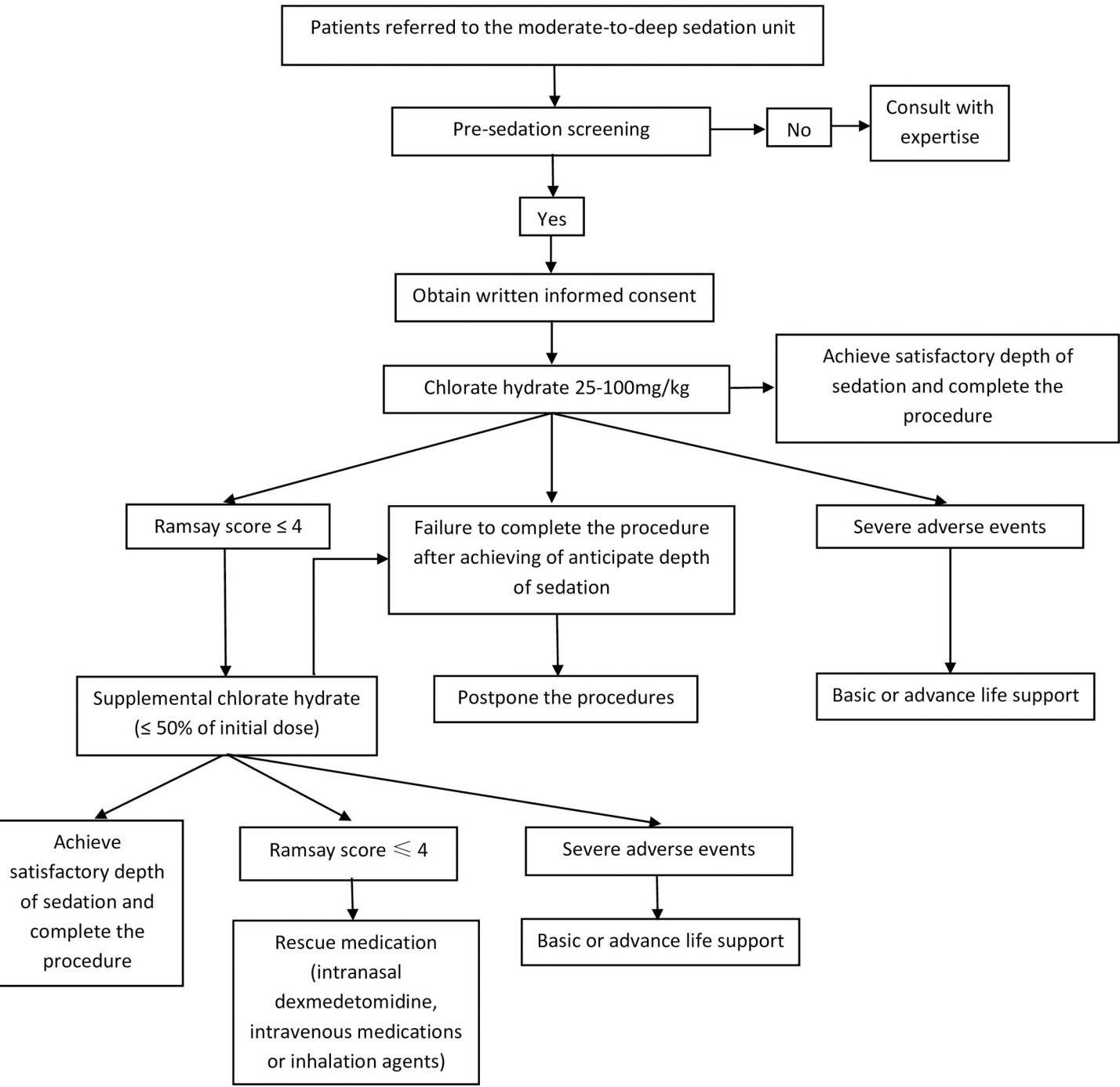

**Fig 1. The process of non-invasive procedural sedation.**

25–100 mg/kg is chosen as the initial dose. To decrease sedative failure during the procedure, a supplemental chloral hydrate with a interval more than 30 min is permitted ($\leq$ 50% of original dose). When satisfactory depth of sedation cannot be achieved, the rescue medications (i.e., intranasal dexmedetomidine, intravenous medications or inhalation agents) are permitted at discretion of the anesthesiologists (Fig 1).

**Table 1. Ramsay scale.**

| Level/score | Clinical description |
|---|---|
| I | Anxious and agitated |
| II | Cooperative, oriented, tranquil |
| III | Responds only to verbal commands |
| IV | Asleep with brisk response to light stimulation |
| V | Asleep with sluggish response to stimulation |
| VI | Asleep without response to stimulation |

Chloral hydrate is administered orally or rectal, and then a portable oxygen saturation monitor is connected after induction of sleep. Satisfactory depth of sedation is defined as Ramsay score (Table 1) > 4 within 30 min. When satisfactory depth of sedation is achieved, the patients are transferred to procedure room by one of sedative nurses. Once the diagnostic procedure is completed, the patients are escorted to recovery room of the sedation unit by the sedative nurse. Then, the patients are kept warm and monitored until fully awake.

## Definitions

Sedation failure with initial dose is defined as follows: ① After initial dose of chloral hydrate, the patients cannot achieve satisfactory depth of sedation within 30 min; ② Failure to complete the diagnostic procedure even after achievement of the anticipated depth of sedation [5]; ③ The occurrence of severe adverse events which prevents the diagnostic procedures continuing.

Sedation failure with supplemental dose was defined as follows: ① After supplemental dosing with chloral hydrate, the patients could not achieve satisfactory depth of sedation within 30 min;② Failure to complete the diagnostic procedure even after achievement of the anticipated depth of sedation [5]; ③ The occurrence of severe adverse events which prevents the diagnostic procedures continuing; ④ The need of other medications in addition to chloral hydrate.

The length of sedation time is defined as the time from drug administration to discharge from sedation unit. Only the patients achieved satisfactory sedation level and completed procedures are calculated.

Minor adverse events contained: (1) vomiting; (2) mild upper airway obstruction which could be improved by changing the posture; (3) delayed awakening (defined as a sedation time > 2h); (4) rash.

Major adverse events were defined as any one of the following events: (1) respiratory depression needs airway management (including mask ventilation, laryngeal mask ventilation, tracheal intubation, or placement of other airway device; (2) laryngospasm or bronchospasm; (3) cardiac arrest; (4) death.

## Outcome measures

The primary outcome was the failure rate of sedation with the initial dose. The secondary outcomes included the failure rate of sedation after supplementation of chloral hydrate, the incidence of major and minor adverse events, initial and supplemental dose of chloral hydrate, and the length of sedation time.

## Statistical analysis

The categorical variables were expressed as percentages. Continuous variables are presented as the mean ± standard deviation [SD] or median and interquartile range [IQR] (25%-75%) if

nonnormally distributed. The student t test was used to compare normally distribute data, otherwise the Mann Whitney U-test was used to compare two groups. The chi-squared test or fisher exact test was used for categorical data, as appropriate. To make the patients with sleep deprivation and those without sleep deprivation comparable, age, gender, weight, type of patients, sedation history, and type of procedures were matched with propensity score method (PSM). Patients were undergone a 1: 1 nearest neighbor matching algorithm without replacement. All data analyses were performed using the R studio version 3.5.2. P < 0.05 was statistically significant, and all tests were two-sided.

## Results

A total of 8415 subjects referred to our moderate-to-deep sedation unit from December 1, 2019 to June 31, 2020. Of those, 178 neonates were excluded. Thirty-five patients were excluded because of incomplete data. Besides, 413 cases were excluded because the initial or supplemental sedatives were not chlorate hydrate, or the guardians were incooperative (Fig 2). Finally, 7789 patients met the eligible criteria were included in our study. Sleep deprivation were performed in 6352 patients (81.6%), which showed a high compliance from the guardians.

All 7789 of the remaining subjects, aged 1 month-13 years, were enrolled in this study. Demographics and sedation characteristics of the eligible subjects were presented in Table 2. Most patients who were sedated were $\leq$ 3 years old (87.4%). The range of ASA level was similar in both groups. There were slightly more boys than girls. The subjects were predominantly outpatients, with a ratio of 66.6%. A list of all procedures performed on patients in this cohort were also presented in Table 2.

## The failure rate of sedation with initial and supplemental chlorate hydrate

The failure rate of sedation with initial chloral hydrate was 8.2% (n = 635), with 483 (7.6%) subjects in sleep deprivation group and 152 (10.5%) in non-sleep deprivation group. The difference was statistically significant (P<0.01). No statistic difference was found in the dosage of the initial chloral hydrate between the two groups (Table 2).

The failure rate of sedation with supplemental chloral hydrate was 0.6% (n = 49), with 36 (0.6%) cases in sleep deprivation group and 13 (0.9%) in non-sleep deprivation group (P = 0.78). The mean of supplemental dose of chloral hydrate was 24 ± 13 mg/kg in sleep deprivation group compared with 24 ± 12 mg/kg in non-sleep deprivation group. Finally, there were 7740 subjects who achieved the depth of sedation and completed the procedures (7154 subjects with the first dose of chlorate hydrate, 586 subjects with second dose of chloral hydrate). Median the length of sedation time for those patients was 58 min (IQR: 45-75 min) (Table 2). No difference was noted in the length of sedation time regardless of whether they were sleep deprived [58 (45, 75) mins] or not [58 (45, 75) mins] (P = 0.95).

Before matching, the baselines of two groups were unbalanced in age, weight, the type of patients, sedation history, and type of procedures. After performing the propensity score matching technique, 1437 pairs were produced, and covariates were well balanced. The failure rate of sedation with initial or second chlorate hydrate, the dose of chloral hydrate, and sedation time were not different between two groups (Table 3).

## Minor and major adverse events

A total of 453 cases (5.8%) had minor adverse events, including 192 cases (0.3%) of PONV, 1 case (0.01%) of allergy, 2 cases (0.03%) of mild upper airway obstruction that can be improved by changing posture, and 258 cases (3.3%) of delayed awakening. Due to respiratory

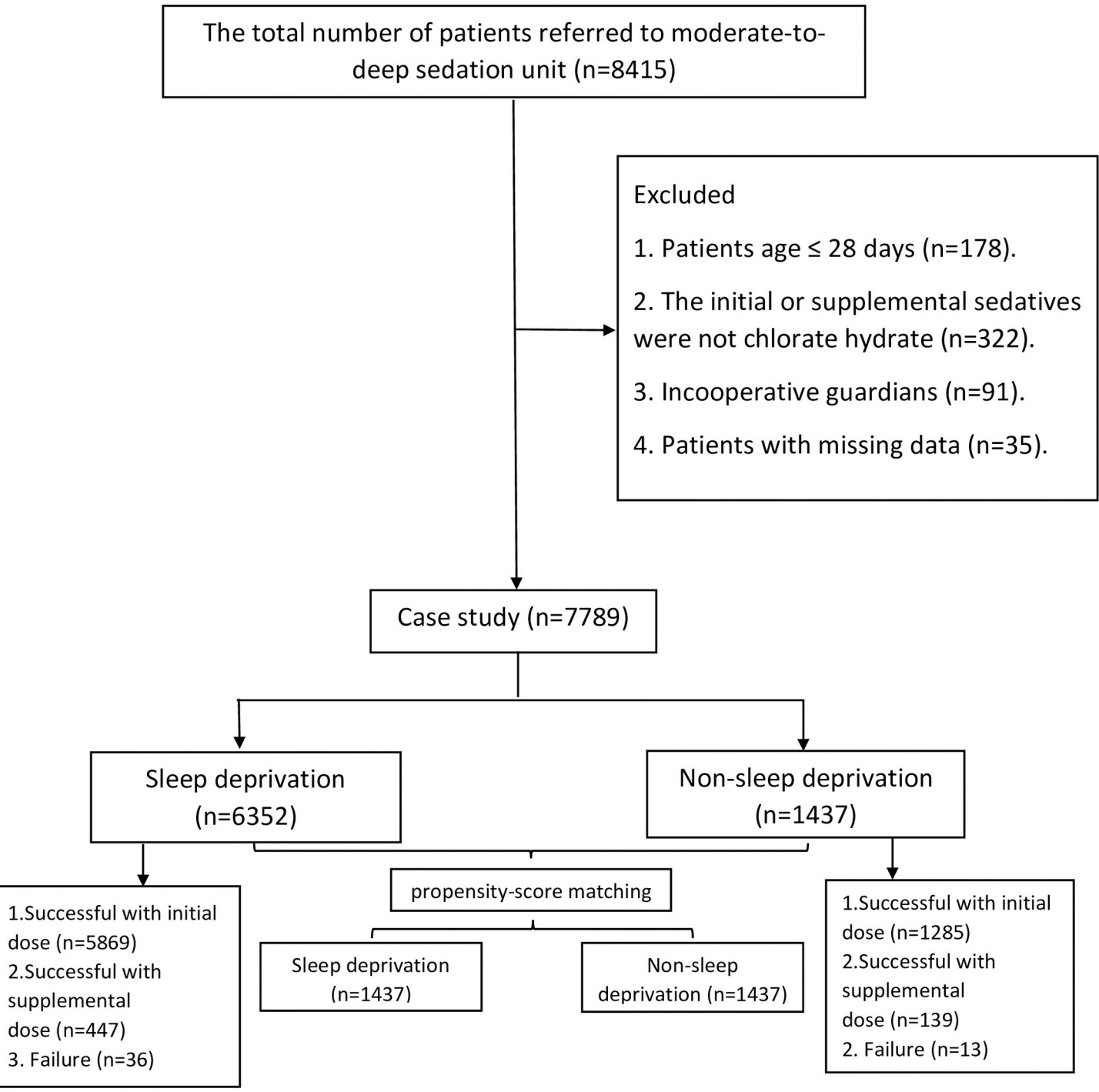

**Fig 2. CONSORT flow chart.**

depression, five patients (0.1%) had to undergo emergency airway management (mask ventilation), which was considered the major adverse events. (Table 4) Notably, before matching, the incidence of vomiting in the sleep deprivation patients was lower than that in the non-sleep deprivation patients (2.1% [131/ 6352 vs. 4.2% [61/1437]; p<0.01). After matching, no difference was detected on the incidence of vomiting (2.1% [42/1437 vs. 4.2% [61/1437]; p = 0.07).

**Table 2. Demographics and sedation characteristics.**

| Characteristics | Total (n = 7789) | Sleep deprivation (n = 6352) | Non-sleep deprivation (n = 1437) | P |
|---|---|---|---|---|
| Gender (M) (n, %) | 4628 (59.4) | 3754 (59.1) | 897 (60.8) | 0.43 |
| Age (n, %) | | | | <0.01* |
| 1–12 months | 4349 (55.8) | 3606 (56.8) | 743 (51.7) | |
| 1–3 years | 2461 (31.6) | 1951 (30.7) | 510(35.5) | |
| 3–6 years | 942 (12.1) | 762 (12.0) | 180 (12.5) | |
| >6 years | 37 (0.4) | 33 (0.5) | 4 (0.3) | |
| Weight (Kg) (Mean ± SD) | 9.6 ± 4.4 | 9.6 ± 4.4 | 9.9 ± 4.4 | 0.03* |
| ASA (n, %) | | | | 0.06 |
| I | 7105 (91.2) | 5815 (91.5) | 1290 (89.8) | |
| II | 632 (8.1) | 499 (7.8) | 133 (9.3) | |
| III | 52 (0.7) | 38 (0.6) | 14 (1.0) | |
| IV | 0 (0.0) | 0 (0.0) | 0 (0.0) | |
| Type of patients (n, %) | | | | |
| Inpatients | 2602 (33.4) | 2027 (31.9) | 575 (40) | <0.01* |
| Outpatients | 5187 (66.6) | 4325 (68.1) | 862 (60) | |
| Sedation history (n, %) | 2612 (33.5) | 2081 (32.8) | 537 (37.0) | <0.01* |
| Sedation failure history (n, %) | 9 (0.1) | 8 (0.1) | 1 (0.1) | 0.89 |
| Type of procedures (n, %)) | | | | <0.01* |
| MRI | 1426 (18.3) | 1167 (18.4) | 259 (18.2) | |
| CT | 676 (8.7) | 505 (8.0) | 171 (11.9) | |
| Lung function | 1754 (22.5) | 1355 (21.3) | 399 (27.8) | |
| Cardiac ultrasound | 1806 (23.2) | 1444 (22.7) | 362 (25.2) | |
| ABR | 1772 (22.8) | 1599 (25.2) | 173 (12.0) | |
| AEP | 121 (15.5) | 90 (1.4) | 31 (2.2) | |
| Others | 7 (0.1) | 7 (0.1) | 0 (0.0) | |
| Performed two procedures | 222 (2.9) | 180 (2.8) | 42 (2.9) | |
| Performed three procedures | 5 (0.1) | 5 (0.1) | 0 (0.0) | |
| Sedation failure with initial dose (n, %) | 635 (8.2) | 483 (7.6) | 152 (10.5) | <0.01* |
| The dosage of initial dose (mg/kg) (Mean ± SD) | 40 ± 13 | 40 ± 13 | 40 ± 13 | 0.47 |
| Sedation failure with supplemental dose (n, %) | 49 (0.6) | 36 (0.6) | 13 (0.9) | 0.78 |
| The dosage of supplemental dose (mg/kg) (Mean ± SD) | 24 ± 12 | 24 ± 13 | 24 ± 12 | 0.97 |
| The length of sedation time # (min) [Median (IQR)] | 58 (45,75) | 58 (45,75) | 58 (45,75) | 0.95 |

ASA: American Society of Anesthesiology; M: Male; MRI: Magnetic resonance; CT: Computer Tomography; ABR: Auditory brain response testing; AEP: Auditory evoked potential.

# The length of sedation time was defined as the sedation time was defined as the time from drug administration to discharge from sedation unit. Only the patients achieved satisfactory sedation level and completed procedures were calculated (n = 7740).

* P<0.05.

## Discussion

In this retrospective cohort study, prior to matching, our results supported that sleep deprivation could decrease the failure rate of chloral hydrate sedation with initial dose. After matching, we found no differences in the failure rate of sedation with initial or supplemental chloral hydrate, no difference in the dose of initial or supplemental chloral hydrate, and no difference in the length of sedation time between the two groups. Our finding indicated that sleep deprivation would not decrease the failure rate of chloral hydrate sedation for non-invasive procedure sedation in children.

**Table 3. Characteristics of sleep deprivation and eligible non-sleep deprivation patients after matching.**

| Characteristics | Propensity score matching | | Exact P value* |
|---|---|---|---|
| | Sleep deprivation(n = 1437) | Non-sleep deprivation(n = 1437) | |
| Gender (M) (n, %) | 901 (62.7) | 874 (60.8) | 0.32 |
| Age (n, %) | | | |
| 1–12 months | 748 (52.1) | 743 (51.7) | 1 |
| 1–3 years | 509 (35.4) | 510 (35.5) | |
| 3–6 years | 176 (12.2) | 180 (12.5) | |
| >6 years | 4 (0.3) | 4 (0.3) | |
| Weight (Kg) (Mean ± SD) | 9.8 ± 4.4 | 9.9 ± 4.4 | 0.64 |
| Type of patients (n, %) | | | 0.47 |
| Inpatients | 555 (38.6) | 575 (40.0) | |
| Outpatients | 882 (61.4) | 862 (60.0) | |
| Sedation history (n, %) | 518 (36.0) | 531 (37.0) | 0.64 |
| Type of procedures (n, %)) | | | 0.91 |
| MRI | 246 (17.1) | 259 (18.0) | |
| CT | 168 (11.7) | 171 (11.9) | |
| Lung function | 423 (29.4) | 399 (27.8) | |
| Cardiac ultrasound | 364 (25.3) | 362 (25.2) | |
| ABR | 173 (12.0) | 173 (12.0) | |
| AEP | 24 (1.7) | 31 (2.0) | |
| Others | 0 (0) | 0 (0) | |
| Performed two procedures | 39 (2.7) | 42 (2.9) | |
| Performed three procedures | 0 (0) | 0 (0) | |
| Sedation failure with initial dose (n, %) | 123 (8.6) | 152 (10.6) | 0.08 |
| The dosage of initial dose (mg/kg) (Mean ± SD) | 40 ± 13 | 40 ± 13 | 0.73 |
| Sedation failure with additional dose (n, %) | 12 (0.8) | 13 (0.9) | 1 |
| The dosage of additional dose (mg/kg) (Mean ± SD) | 24 ± 11 | 27 ± 10 | 0.43 |
| The length of sedation time # (min) [Median (IQR)] | 58 (45, 75) | 58 (45, 75) | 0.93 |

M: Male; MRI: Magnetic resonance; CT: Computer Tomography; ABR: Auditory brain response testing; AEP: Auditory evoked potential.

# The length of sedation time was defined as the sedation time was defined as the time from drug administration to discharge from sedation unit. Only the patients achieved satisfactory sedation level and completed procedures were calculated (n = 2849).

To reduce the need for sedation in children and adolescents, a simple instruction for partial sleep deprivation prior to the electroencephalogram was published in 2004. The authors presented that when the partial sleep deprivation was implemented, the proportion of patients undergoing the test fell asleep without sedation was increased, from 19% to 55% [10]. As Cynthia et al. mentioned a corollary to the concept that sleep deprivation enhanced sedation success was the notion that it might reduce the requirement of drug necessary for a successful sedation [9]. In other words, it might reduce the dose of drug for a successful sedation or improve first-time success rate. Based on above theories, mild sleep deprivation was encouraged prior to the procedure in our unit, but it was a tough task for guardians, especially when their children were<3 years of age. Planned comparison revealed that the sedation failure rate with initial dose of chloral hydrate in sleep deprivation group was lower that non-sleep deprivation (sleep deprivation: 7.6% vs. non-sleep deprivation: 10.5%; P<0.01), which seemed to suggest that patients with sleep deprivation had a higher first-time success rate. Those results seemed to support the theory that sleep deprivation had some beneficial effects in reducing the pediatric sedation first-time failure rate. However, the baseline of patients in two groups were

**Table 4. Minor and major adverse events.**

| Minor adverse events | Before Matching | | | | After Matching | | |
| --- | --- | --- | --- | --- | --- | --- | --- |
| | Total number (n = 7789) | Sleep deprivation (n = 6352) | Non-sleep deprivation (n = 1437) | P value | Sleep deprivation (n = 1437) | Non-sleep deprivation (n = 1437) | Adjusted P value |
| Vomiting (n, %) | 192 (2.5) | 131 (2.1) | 61 (4.2) | <0.01* | 42 (2.9) | 61 (4.2) | 0.07 |
| Rash (n, %) | 1 (0.01) | 0 (0.0) | 1 (0.07) | 0.42 | 0 (0.0) | 1 (0.07) | 0.42 |
| Mild upper airway obstruction (n, %) | 2 (0.03) | 2 (0.03) | 0 (0.0) | 1 | 0 (0.0) | 0 (0.0) | NA |
| Delayed awakening (n, %) # | 258 (3.3) | 213 (3.4) | 45 (3.1) | 0.7 | 45 (3.1) | 45 (3.1) | 1 |
| **Major adverse events** | | | | | | | |
| Respiratory depression (n, %) | 5 (0.1) | 4 (0.06) | 1 (0.07) | 1 | 1 (0.07) | 1 (0.07) | 1 |
| Laryngospasm or bronchospasm (n, %) | 0 (0.0) | 0 (0.0) | 0 (0.0) | NA | 0 (0.0) | 0 (0.0) | NA |
| Cardiac arrest (n, %) | 0 (0.0) | 0 (0.0) | 0 (0.0) | NA | 0 (0.0) | 0 (0.0) | NA |
| Death (n, %) | 0 (0.0) | 0 (0.0) | 0 (0.0) | NA | 0 (0.0) | 0 (0.0) | NA |

# Only the patients achieved satisfactory sedation level and completed procedures were calculated (n = 7772). Of those, 6339 patients were in sleep deprivation group and 1433 patients were in non-sleep deprivation group.

* P<0.05.

different. Compared to non-sleep deprivation patients, sleep deprivation patients were more likely to be younger than 1 year old and have lower body weight. A higher percentage of non-sleep deprived patients had a history of sedation and a higher percentage were inpatients. Type of procedures were also varied greatly. As far as we know, the age, weight, and type of procedures played a vital role in determining the sedation endpoint. After propensity-score matching, 1437 pairs were produced, and the basic characteristics of enrolled patients were well balanced. It was worth noting that after propensity-matching, the sedation failure rate with initial dose was 8.6% for sleep deprivation patients versus 10.6% for non-sleep deprivation patients, showing an upward trend, but this was not a statistically significantly difference. Based on current evidence, the assumption about sleep deprivation facilitating chloral hydrate sedation effects was not supported, which was in agreement with Cynthia's study [9].

In 2017, Mataftsi et al. had summarized existing literature about chloral hydrate for procedural sedation in pediatric ophthalmology. The authors reported that efficacy in achieving satisfied sedation using chloral hydrate with a first dose of 60–100 mg/kg was 88% - 99% [11]. Similarly, a retrospective review conducted in 2014 found that chloral hydrate sedation was successful in 94.2% of cases with a single mean dose of 77.5 mg/kg [12]. In our setting, for safety, sedation practitioners were encouraged to prescribe sedatives starting from a small dose, and supplemental dose was permitted based the sedation practitioners' assessment. Interestingly, the mean initial dose of chloral hydrate was about 40 mg/kg with a success rate about 90%. And the overall success rate could reach up to 99% with a supplemental dose (mean dose: 24 mg/kg). Our results suggested that in the majority of children, even a lower dose of chloral hydrate could achieve satisfied sedation, which was in line the conception that using the minimum dose of sedative to achieve maximum effects. However, the eligible subjects in our study underwent non-invasive rather than invasive procedures, which might explain the high success rate of sedation with less dose of chloral hydrate. Moreover, most previous studies enrolled the patients with age > 1year. In our study, children under 1 year old accounted for more than 50%. Despite chloral hydrate had a wide margin of safety, minimum effective dose was still unknown. One study recommended that the dose for chloral hydrate was 50 mg/kg under the

age of 2 years and 75 mg/kg over the age of 2 years [4], while another suggested children younger than 6 months received a target dose of 50 mg/kg, and all others received a target dose of 100 mg/kg [13]. There was no known the dosage threshold in different ages. We strongly suspected that children of different ages had different minimum effective dose on chloral hydrate sedation, but this was our assumption and further high-quality evidence focusing on different ages was needed to clarify it.

Additionally, we evaluated the effects of sleep deprivation on the sedation time. A complete data was available in 7740 patients who finished the non-invasive procedure with chloral hydrate sedation. We discovered that the length of sedation time was similar for both sleep deprivation [58 (45,75) mins] and non-sleep deprived [58 (45,75) mins] patients (P = 0.95). Although matching was performed and 2867 patients were available, no difference was detected. In contrary, previous retrospective studies had suggested that sleep deprivation patients required significantly more nursing care hours than non-sleep deprivation patients [9]. The possible explanation for this finding was that the beds in the recovery room was limited, resulting in a very efficient process. When the procedure was complete, the sedation practitioners might awake the child from sedation, and the length of sedation time might be influenced by the extent of such stimulation.

From a historical perspective, chloral hydrate was a reliable sedative with a high efficacy and safety, which had been extensive studied [11]. A meta-analysis that assessed available randomized controlled trials (RCTs) suggested that the use of chloral hydrate for procedure sedation was safe with appropriate monitoring. The results reported that compared to other sedatives, chloral hydrate had a higher OR (3.49, 95% CI 1.32 to 9.21) for successful sedation without raising adverse events [11]. We also found that chloral hydrate could be safely used in pediatric patients for painless procedural sedation. Most reported adverse events were not serious and transient, i.e., vomiting, delayed awaking, rash, and mild upper airway obstruction. The severe adverse events encountered in our study were respiratory depression which had been well resolved without complications. Interestingly, before matching, the incidence of vomiting was higher in sleep deprivation than non-sleep deprivation patients, and after matching, this difference disappeared. This difference was difficult to explain and its clinical significance is limited. Furthermore, chloral hydrate was no longer used in some countries (e.g., France, Italy) because of potential carcinogenicity concern. In fact, the direct relationship between chloral hydrate and carcinogenicity was unknown. Most of the available literature referring to chloral hydrate on carcinogenicity was long-term abuse. Caldwell et al. reported that chloral hydrate feeding 2 years increased the incidences of hepatocellular adenoma [14]. This was a very extreme design, rarely seen in clinical practice. Therefore, their results could not be simply interpreted into the possible harm caused by chloral hydrate as a sedative. Further research was needed to discuss the carcinogenic effects of chloral hydrate on pediatric patients requiring repeated sedation.

There were several limitations in our study. First, the implementation of sleep deprivation required the cooperation of parents. To keep a child awake, at least one or even two parents must be awake. This effort could not be evaluated. Next, this was a retrospective study, and selection bias might exist in the process. For example, because sleep deprivation might cause the child to cry and disturb others' rest, it was more difficult to be implemented in a multiple-bed ward. This was why there were fewer inpatients in the sleep deprivation group compared to the non-sleep deprivation group. Additionally, in the current study, the age ranged from 1 month to 13 years, which was a very wide. To get a more convincing result, judgements on the impact of sleep deprivation might need to be made by certain more restricted age groups. In the future, the patients should be stratified by age to assure directness of evidence. Last,

different type of procedures had the different sedation endpoint. To reduce bias on type of procedures, the propensity-score matching was conducted to balance the baseline of two groups.

## Conclusions

The current results do not support that sleep deprivation had a beneficial effect in reducing the pediatric chloral hydrate sedation failure rate. The routine use of sleep deprivation for pediatric sedation is unnecessary. A lower dose of chloral hydrate may achieve satisfactory sedation for patients undergoing noninvasive procedures.

## Supporting information

**S1 Checklist. STROBE statement—Checklist of items that should be included in reports of *cohort studies*.**
(DOCX)

## Author Contributions

**Conceptualization:** Yu Cui, Langtao Guo.

**Data curation:** Yu Cui, Langtao Guo, Qixia Mu, Qin Cheng, Lu Kang, Yani He, Min Tang, Qunying Wu.

**Investigation:** Yu Cui.

**Supervision:** Yu Cui.

**Validation:** Yu Cui.

**Writing – original draft:** Yu Cui, Langtao Guo.

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
