## [Decision Letter · Decision Letter 0]

11 Dec 2020

PONE-D-20-26402

Sleep deprivation did not enhance the success rate of chloral hydrate sedation for non-invasive procedural sedation in pediatric patients

PLOS ONE

Dear Dr. Cui,

Thank you for submitting your manuscript to PLOS ONE. After careful consideration, we feel that it has merit but does not fully meet PLOS ONE’s publication criteria as it currently stands. Therefore, we invite you to submit a revised version of the manuscript that addresses the points raised during the review process.

In the current observational study, Yu Cui et.al assessed the association between mild sleep deprivation and sedative effects in non-invasive procedures. They concluded that sleep deprivation has no effect on reducing the failure rate of chloral hydrate sedation in children. 

The manuscript has been assessed by two reviewers and their comments are available below. The reviewers concerned more details regarding the results. Overall, there were too many confounding factors in the statistical comparison and subgroup analysis was necessary to control bias.

We look forward to receiving your revised manuscript.

Kind regards,

JianJun Yang, M.D., Ph.D.

Academic Editor

PLOS ONE

Journal Requirements:

2. For more information on PLOS ONE's expectations for statistical reporting, please see https://journals.plos.org/plosone/s/submission-guidelines.#loc-statistical-reporting. Please update your Methods and Results sections accordingly.

3.Thank you for stating the following financial disclosure:

 [NO].

Reviewers' comments:

Reviewer's Responses to Questions

**Comments to the Author**

1. Is the manuscript technically sound, and do the data support the conclusions?

Reviewer #1: Yes

Reviewer #2: Yes

2. Has the statistical analysis been performed appropriately and rigorously? 

Reviewer #1: Yes

Reviewer #2: Yes

3. Have the authors made all data underlying the findings in their manuscript fully available?

Reviewer #1: Yes

Reviewer #2: Yes

4. Is the manuscript presented in an intelligible fashion and written in standard English?

Reviewer #1: Yes

Reviewer #2: No

5. Review Comments to the Author

Reviewer #1: Please clarify definition of "clear fluids" giving examples

Please review the fasting guidelines for formula milk. It should be 6 hours.

Please clarify "Ramsay Score". Better in a separate table.

Otherwise the manuscript is well prepared and written.

Reviewer #2: This is an interesting study investigating the impact of sleep deprivation on the success rate of chloral hydrate sedation. The topic is of interest, because sleep deprivation presents a feasible measure without bearing the risk for relevant adverse side effects. I have a few comments to make.

The English language and grammar need to be revised throughout the manuscript. (e.g. Page 5, line 87: replace polled with pooled)

Control group: how was the control group comprised? Where these the patients excluded from the sleep deprivation group for reasons such as other sedatives, uncooperative guardians, neonates and incomplete data? From where were these controls taken, if all parents were consistently advised to perform sleep deprivation or if sleep deprivation is always attempted? Is the control group comprised of patients where sleep deprivation failed for various reasons? If so, then that would be a problematic comparison with risk of bias.

The age range from 1 month to 13 years is a very wide, considering the outcome of interest. Although they may all be considered pediatric, I would expect significant differences in the physiology of sedation and awareness between a 13year old and a 1month old child given the heterogeneity among different age groups. In the current study >50% were children under the age of 1 year. This could to some extent explain the lack of effect and the discrepancies in current literature. Moreover, judgements on the impact of sleep deprivation may need to be made by certain more restricted age groups. The authors should address this in the limitations section and mention that future studies should stratify by age to assure directness of evidence.

The mix of inpatients and outpatients is another limitation that should be more clearly addressed because sleep deprivation in a home setting may be very different from sleep deprivation in a hospital environment.

The current study question would largely benefit from a prospective standardized setting.

Can the authors discuss current literature on effect of sleep deprivation with the use of other sedative medications than chloral hydrate? The effect of sleep deprivation may be different with the use of other sedative agents.

As mentioned by the authors, the type of procedure is also an important source of bias given the different sedation endpoints.

6. PLOS authors have the option to publish the peer review history of their article (what does this mean?). If published, this will include your full peer review and any attached files.

Reviewer #1: **Yes: **Sherif S Sultan

Reviewer #2: No

---

## [Author Response · Author response to Decision Letter 0]

12 Dec 2020

Journal Requirements:

Answer: Thanks. We have revised the manuscript according to PlOS ONE’s style requirements.

2. For more information on PLOS ONE's expectations for statistical reporting, please see https://journals.plos.org/plosone/s/submission-guidelines.#loc-statistical-reporting. Please update your Methods and Results sections accordingly.

Answer: Thanks. We have updated our methods and results sections accordingly.

3.Thank you for stating the following financial disclosure:

 [NO].

Please clarify the sources of funding (financial or material support) for your study. List the grants or organizations that supported your study, including funding received from your institution.

State what role the funders took in the study. If the funders had no role in your study, please state: “The funders had no role in study design, data collection and analysis, decision to publish, or preparation of the manuscript.”

If any authors received a salary from any of your funders, please state which authors and which funders.

If you did not receive any funding for this study, please state: “The authors received no specific funding for this work.”

Answer: We have clarified that this research did not receive any specific grant from funding agencies in the public, commercial, or not-for-profit sectors. (Page 20, line 355-356)

4. Please include captions for your Supporting Information files at the end of your manuscript, and update any in-text citations to match accordingly. Please see our Supporting Information guidelines for more information: http://journals.plos.org/plosone/s/supporting-information.Reviewers' comments:

Reviewer's Responses to Questions

Answer: We don’t have any supporting Information files.

Comments to the Author

1. Is the manuscript technically sound, and do the data support the conclusions?

Reviewer #1: Yes

Reviewer #2: Yes

Answer: Thanks. 

2. Has the statistical analysis been performed appropriately and rigorously?

Reviewer #1: Yes

Reviewer #2: Yes

Answer: Thanks. 

3. Have the authors made all data underlying the findings in their manuscript fully available?

Reviewer #1: Yes

Reviewer #2: Yes

Answer: Thanks. 

4. Is the manuscript presented in an intelligible fashion and written in standard English?

Reviewer #1: Yes

Reviewer #2: No

Answer: The English language and grammar have been revised throughout the manuscript. 

5. Review Comments to the Author

Answer: We have used the space provided to explain our answers to the reviewer’s comments.

Reviewer #1: Please clarify definition of "clear fluids" giving examples

Answer: We have clarified the definition of “clear fluids” as follows.

“The clear liquids were referred to water, fruit juices without pulp, or carbohydrate-containing fluids without protein.” (Page 7, Line 115)

Please review the fasting guidelines for formula milk. It should be 6 hours.

Answer: Yes, in general anesthesia, the fasting guidelines for formula milk should be 6 hours. However, in procedural sedation, no consensus was achieved [1]. Although sedation and anesthesia are a continuum, it is not clear that the same set of fasting intervals should necessarily be equally applicable to all sedation depths, sedation durations, procedure types and patient conditions or comorbidities. Procedural sedation intentionally targets a state in which protective airway reflexes are retained, while general anesthesia denotes a state in which they are absent. In a retrospective study enrolled 17948 pediatric patients undergoing procedural sedation, the fasting requirement was 1h. No aspiration was reported in that study [2]. Therefore, the patients were fasted from clear liquids for 2h and from formula, breast milk for 4h per our institutional guidelines.

References:

1. Green SM, Leroy PL, Roback MG, et al. An international multidisciplinary consensus statement on fasting before procedural sedation in adults and children. Anaesthesia. 2020;75:374-385. doi:10.1111/anae.14892

2. Yang F, Liu Y, Yu Q, et al. Analysis of 17 948 pediatric patients undergoing procedural sedation with a combination of intranasal dexmedetomidine and ketamine. Paediatr Anaesth. 2019;29(1):85-91. doi:10.1111/pan.13526

Please clarify "Ramsay Score". Better in a separate table.

Answer: This is a valuable suggestion. We have clarified "Ramsay Score" in a separate table. (Table 1)

Otherwise the manuscript is well prepared and written. 

Answer: Thanks.

Reviewer #2: This is an interesting study investigating the impact of sleep deprivation on the success rate of chloral hydrate sedation. The topic is of interest, because sleep deprivation presents a feasible measure without bearing the risk for relevant adverse side effects. I have a few comments to make.

The English language and grammar need to be revised throughout the manuscript. (e.g. Page 5, line 87: replace polled with pooled)

Answer: Thank you for your valuable comments. The English language and grammar have been revised throughout the manuscript.

Control group: how was the control group comprised? Where these the patients excluded from the sleep deprivation group for reasons such as other sedatives, uncooperative guardians, neonates and incomplete data? From where were these controls taken, if all parents were consistently advised to perform sleep deprivation or if sleep deprivation is always attempted? Is the control group comprised of patients where sleep deprivation failed for various reasons? If so, then that would be a problematic comparison with risk of bias.

Answer: The eligible criteria were as follows: (1) children who underwent drug assistant sedation for non-invasive procedures in our sedation center; (2) pediatric patients who received choral hydrate as the initial and supplemental sedative. It was indicated that only the patients who received choral hydrate as the initial and supplemental sedative were enrolled in our study. 

For appointed patients, sleep deprivation was encouraged in our institution, but some parents did not follow our advice. For example, because sleep deprivation might cause the child to cry and disturb others’ rest, it was more difficult to be implemented in a multiple-bed ward. And for non-appointed patients, sleep deprivation could not be achieved. Therefore, when the patients arrived at the sedation unit, the guardians were asked in a nonjudgmental question (e.g., How many hours of sleep their child received yesterday? And how many hours of sleep their child usually received?). Then, the answers were recorded. In our study, any child who sleeps more than 2 hours less than usual is considered sleep deprived.

The age range from 1 month to 13 years is a very wide, considering the outcome of interest. Although they may all be considered pediatric, I would expect significant differences in the physiology of sedation and awareness between a 13year old and a 1month old child given the heterogeneity among different age groups. In the current study >50% were children under the age of 1 year. This could to some extent explain the lack of effect and the discrepancies in current literature. Moreover, judgements on the impact of sleep deprivation may need to be made by certain more restricted age groups. The authors should address this in the limitations section and mention that future studies should stratify by age to assure directness of evidence.

Answer: This is a valuable question. We have expanded our limitations.

“Additionally, in the current study, the age ranged from 1 month to 13 years, which was a very wide. To get a more convincing result, judgements on the impact of sleep deprivation might need to be made by certain more restricted age groups. In the future, the patients should be stratified by age to assure directness of evidence.” (Page 20, line 343-346)

The mix of inpatients and outpatients is another limitation that should be more clearly addressed because sleep deprivation in a home setting may be very different from sleep deprivation in a hospital environment.

Answer: Yes. We agreed with you. We have addressed this in the limitations.

“For example, because sleep deprivation might cause the child to cry and disturb others’ rest, it was more difficult to be implemented in a multiple-bed ward. This was why there were fewer inpatients in the sleep deprivation group compared to the non-sleep deprivation group.” (Page 19, line 340)

The current study question would largely benefit from a prospective standardized setting.

Answer: Yes. We agreed with you. In the future, a prospective randomized controlled trial will be conducted in our setting.

Can the authors discuss current literature on effect of sleep deprivation with the use of other sedative medications than chloral hydrate? The effect of sleep deprivation may be different with the use of other sedative agents.

Answer: We are sorry about that the effect of sleep deprivation with the use of other sedative medications has not been well summarized. However, we intent to analyze the association between mild sleep deprivation and sedative effects in patients undergoing procedural sedation with dexmedetomidine in the following months.

As mentioned by the authors, the type of procedure is also an important source of bias given the different sedation endpoints.

Answer: Yes. We have mentioned it in the discussion section.

6. PLOS authors have the option to publish the peer review history of their article (what does this mean?). If published, this will include your full peer review and any attached files.

Do you want your identity to be public for this peer review? For information about this choice, including consent withdrawal, please see our Privacy Policy.

Reviewer #1: Yes: Sherif S Sultan

Reviewer #2: No

Answer: Thanks again to all the reviewers and editors.

---

## [Decision Letter · Decision Letter 1]

29 Dec 2020

Sleep deprivation did not enhance the success rate of chloral hydrate sedation for non-invasive procedural sedation in pediatric patients

PONE-D-20-26402R1

Dear Dr. Cui,

We’re pleased to inform you that your manuscript has been judged scientifically suitable for publication and will be formally accepted for publication once it meets all outstanding technical requirements.

Kind regards,

JianJun Yang, M.D., Ph.D.

Academic Editor

PLOS ONE

Additional Editor Comments (optional):

Reviewers' comments:

Reviewer's Responses to Questions

**Comments to the Author**

1. If the authors have adequately addressed your comments raised in a previous round of review and you feel that this manuscript is now acceptable for publication, you may indicate that here to bypass the “Comments to the Author” section, enter your conflict of interest statement in the “Confidential to Editor” section, and submit your "Accept" recommendation.

Reviewer #1: All comments have been addressed

Reviewer #2: All comments have been addressed

2. Is the manuscript technically sound, and do the data support the conclusions?

Reviewer #1: Yes

Reviewer #2: Yes

3. Has the statistical analysis been performed appropriately and rigorously? 

Reviewer #1: Yes

Reviewer #2: Yes

4. Have the authors made all data underlying the findings in their manuscript fully available?

Reviewer #1: Yes

Reviewer #2: Yes

5. Is the manuscript presented in an intelligible fashion and written in standard English?

Reviewer #1: Yes

Reviewer #2: Yes

6. Review Comments to the Author

Reviewer #1: Paper is accepted for publishing after latest changes provided by the authors in the file sent by the editorial manager

Reviewer #2: The authors have sufficiently addressed the comments and expanded the limitations section to include the residual limitations.

7. PLOS authors have the option to publish the peer review history of their article (what does this mean?). If published, this will include your full peer review and any attached files.

Reviewer #1: **Yes: **Sherif S Sultan

Reviewer #2: **Yes: **Crispiana Cozowicz

---

## [Editor Report · Acceptance letter]

30 Dec 2020

PONE-D-20-26402R1 

Sleep deprivation did not enhance the success rate of chloral hydrate sedation for non-invasive procedural sedation in pediatric patients 

Dear Dr. Cui:

I'm pleased to inform you that your manuscript has been deemed suitable for publication in PLOS ONE. Congratulations! Your manuscript is now with our production department. 

Kind regards, 

on behalf of

Dr. JianJun Yang 

Academic Editor

PLOS ONE